# Pathomorphological Findings and Infectious Diseases in Selected European Brown Hare (*Lepus europaeus Pallas, 1778*) Populations from Schleswig-Holstein, Germany

**DOI:** 10.3390/pathogens12111317

**Published:** 2023-11-05

**Authors:** Marcus Faehndrich, Benno Woelfing, Jana C. Klink, Marco Roller, Wolfgang Baumgärtner, Peter Wohlsein, Katharina Raue, Christina Strube, Christa Ewers, Ellen Prenger-Berninghoff, Jutta Verspohl, Antonio Lavazza, Lorenzo Capucci, Herbert Tomaso, Ursula Siebert

**Affiliations:** 1Institute for Terrestrial and Aquatic Wildlife Research, University of Veterinary Medicine Hannover, Foundation, 30559 Hannover, Germany; marcus.faehndrich@tiho-hannover.de (M.F.); jana.christina.klink@tiho-hannover.de (J.C.K.); marco.roller@zoo.karlsruhe.de (M.R.); 2Department of Pathology, University of Veterinary Medicine Hannover, Foundation, 30559 Hannover, Germany; wolfgang.baumgaertner@tiho-hannover.de (W.B.); peter.wohlsein@tiho-hannover.de (P.W.); 3Institute for Parasitology, Centre for Infection Medicine, University of Veterinary Medicine Hannover, Foundation, 30559 Hannover, Germanychristina.strube@tiho-hannover.de (C.S.); 4Institute of Hygiene and Infectious Diseases of Animals, Justus Liebig University Giessen, 35390 Giessen, Germany; christa.ewers@vetmed.uni-giessen.de (C.E.); ellen.prenger-berninghoff@vetmed.uni-giessen.de (E.P.-B.); 5Institute for Microbiology, University of Veterinary Medicine Hannover, Foundation, 30559 Hannover, Germany; jutta.verspohl@tiho-hannover.de; 6Istituto Zooprofilattico Sperimentale della Lombardia e dell’Emilia Romagna, 25124 Brescia, Italy; antonio.lavazza@izsler.it (A.L.); lorenzo.capucci@izsler.it (L.C.); 7Friedrich-Loeffler-Institut—Federal Research Institute for Animal Health (FLI), Institute of Bacterial Infections and Zoonoses, 07743 Jena, Germany; herbert.tomaso@fli.de

**Keywords:** *Lepus europaeus*, hepatitis, steatitis, nephritis, EBHSV, *Eimeria* spp., *Trichostrongylus* spp., *Graphidium strigosum*

## Abstract

In the northernmost German federal state Schleswig-Holstein, populations of European brown hares (*Lepus europaeus Pallas, 1778*) show diverse densities and varying courses over the years. To examine differences in pathomorphological findings and infectious diseases as possible reasons for varying population dynamics, we assessed 155 hunted hares from three locations in Schleswig-Holstein from 2016 to 2020. We investigated the association of location, year, age, and sex of animals to certain pathomorphological findings and infectious diseases. Frequent pathomorphological findings were intestinal parasites (63.9%), hepatitis (55.5%), nephritis (31.0%), steatitis (23.2%), enteritis (13.5%), and pneumonia (5.2%). Body condition differed significantly between locations, and the prevalence of pneumonia was significantly higher in females. Enteritis was not detected in 2019, when much more juveniles were sampled. Hepatitis and nephritis occurred significantly more often in 2016 and among adults. Additionally, more adults showed hepatitis with concurrent serotitre for European brown hare syndrome virus (EBHSV), while intestinal parasitosis as well as high excretion rates of coccidia were more common in juveniles. Sampled animals showed high infection rates with *Eimeria* spp. (96.1%), *Trichostrongylus* spp. (52.0%), *Graphidium strigosum* (41.2%), and a high seroprevalence (90.9%) for EBHSV, without severe symptoms. This study revealed a low prevalence of infectious pathogens, but a high prevalence of chronic inflammations of unknown origin in the tested brown hare populations. Overall, our results indicate a rather minor importance of infectious diseases for observed population dynamics of analysed hare populations in Schleswig-Holstein.

## 1. Introduction

The European brown hare (*Lepus* (*L.*) *europaeus Pallas*, *1778*), originally native to European steppes, nowadays inhabits several countries across the globe and is an important small game species [1,2]. For this reason, hares have been under intense monitoring by hunters for many decades and so a decline in hare populations was initially observed in many countries by analysing hunting bag records [3,4,5,6]. In the northernmost German federal state Schleswig-Holstein, the more precise and scientifically evaluated spotlight strip census [7,8,9] has complemented hunting bag data since 1995 in up to 85 different reference areas following Pfister and Rimanthe [10], modified by Pegel [11]. In this area, hare densities are still high (average 2021: 19.5 hares/km^2^ [12]), compared with the German median of 17 hares/km^2^ [13], but also, here hunting bags have been decreasing for several decades and hunters are concerned. The spotlight strip method confirmed a moderate decline in population summer growth [14] and put the *L*. *europaeus* on the Red List of Schleswig-Holstein, classified as near-threatened [15]. Additionally, population densities are quite variable in this federal state and show regular undulating population fluctuations [14] (Figure 1). In general, the coastal areas have higher average densities (marshland: 32.0 hares/km^2^; eastern hills: 17.7 hares/km^2^) compared to the inland-located geest landscape (16.8 hares/km^2^) [12,16].

Next to environmental influences like habitat variability [17,18,19,20], predator pressure [21,22,23,24], intensification of agricultural practice [25,26], and influences of weather or climate [27,28], infectious diseases were discussed to have an impact on population densities of *L*. *europaeus* [29,30,31,32,33]. The latter have already been described for certain regions of Schleswig-Holstein [32,34,35,36,37,38,39,40,41,42] and state-wide investigations have even been conducted in the past and updated lately [43,44]. During all these studies, certain pathomorphological findings were detected regularly, e.g., hepatitis [30,31,32,35,43,45,46,47,48,49,50,51], nephritis [30,31,32,43,45,47,52,53,54], enteritis [30,31,35,43,45,47,53,55], and pneumonia [30,31,32,43,45,47,52,53,55]. Furthermore, intestinal parasites were found regularly [30,31,41,43,44,45,46,47,53,56,57,58]. Mentioned pathomorphological findings were, amongst others, described together with viral and bacterial agents like European brown hare syndrome virus (EBHSV) [49,59], rabbit haemorrhagic disease virus type 2 (RHDV2) [60], *Yersinia* (*Y.*) *pseudotuberculosis (Pfeiffer, 1889)* [29,61], *Pasteurella* (*P.*) *multocida* (*Trevisan, 1887*) [29], and *Francisella* (*F*.) *tularensis (McCoy and Chapin, 1912)* [62,63], as well as parasites like *Eimeria* (*E*.) spp. [61,62,64,65,66] and *Trichostrongylus* (*T*.) *retortaeformis (Zeder, 1800)* [29,61].

The aim of this study was to investigate whether and how infectious diseases and pathomorphological findings in different locations of Schleswig-Holstein are influenced, based on the investigation of hunted animals, to address the potential role of infectious diseases in the different developments of brown hare populations in this German federal state.

## 2. Materials and Methods

### 2.1. Animals and Hunting Grounds

In total, 155 free-ranging *L. europaeus* were collected from three different hunting grounds (Tetenbüll, Friedrichskoog, Elpersbüttel) within the northernmost German federal state of Schleswig-Holstein during the usual hunting season (1 October–31 December) between 2016 and 2020. Therefore, animals were subsampled randomly from the hunting bag of each hunting ground.

Investigated hunting grounds are located in the administrative districts of Dithmarschen and North Frisia and represent the natural region of North Sea marshland [67]. Elpersbüttel (54°07′ N, 9°04′ E) and Friedrichskoog (54° N, 8°91′ E) are located south of the river Eider in the marshland of Dithmarschen (Figure 2). This open cultural landscape is characterised by intensive cultivation of grains, root crops and cabbage, a dense network of drains and very little grassland, hatches, and woodland [68]. Tetenbüll (54°35′ N, 8°82′ E) is located north of the river Eider on the peninsula Eiderstedt, which is a grassland-shaped open cultural landscape with a minor but increasing agriculture, patches of woodland, and a dense network of drains and ponds [69]. Because of the spatial proximity, climatic conditions between the investigated locations can be considered equal.

Selection criteria for hunting grounds were comparable environmental conditions, no possibility of exchange of individuals between hunting grounds, and different population densities. The minimum distance between the closest hunting grounds Elpersbüttel and Friedrichskoog is approximately 12 km, and as the home range of *L. europaeus* is given with a maximum of 300 ha [1], an exchange of animals between the hunting grounds is unlikely. Furthermore, population density needed to be high enough to sample a representative number of individuals over multiple years without a negative impact on local population. Furthermore, a certain density is also needed for spreading of infectious diseases [32,70] for this rather solitary animal [71]. Hare densities of all three hunting grounds were above the average for Schleswig-Holstein (spring 2016: Elpersbüttel: 40.33 hares/km^2^; Friedrichskoog: 25.33 hares/km^2^; Tetenbüll: 43.82 hares/km^2^) but developed quite differently until autumn 2016 with varying summer growth (Elpersbüttel: 38.67 hares/km^2^, −4%; Friedrichskoog: 48.33 hares/km^2^, +91%; Tetenbüll: 24.47 hares/km^2^, −44%) [12].

### 2.2. Necropsy and Histopathology

Necropsies of all carcasses were performed within 48 h postmortem at the Institute for Terrestrial and Aquatic Wildlife Research, University of Veterinary Medicine Hannover, Foundation, Hannover, Germany, followed by histopathological analyses conducted at the Department of Pathology, University of Veterinary Medicine Hannover, Foundation, Hannover, Germany. The methods have been previously described by Faehndrich et al. [43].

Acute haemorrhages, bone fractures, and organ or tissue lacerations were attributed to the method of hunting and are not reported in this study. The remaining pathomorphological findings were categorised into the following groups: alimentary system, cardiovascular system, abdominal and thoracic cavities, haematopoietic and endocrine system, respiratory system, locomotory system, as well as urinary and genital system.

Parasites and parasite stages detected during histopathological analyses are demonstrated and discussed within the pathomorphological findings section and complement the results of parasitological investigations.

Within this study, we focused our analyses on pathomorphological findings, which were detected frequently and might be associated with infectious diseases. Other detected pathomorphological findings are mentioned only in association with these ones.

### 2.3. Parasitology

For parasitological analyses, each animal was first examined macroscopically for endoparasites with a special focus on the gastrointestinal tract. Therefore, every organ was assessed from the outside and sliced to search for intraparenchymal and intraluminal parasites. Parasites found were first stored in water and later in 70% alcohol for preservation. Subsequent analyses and species identification were performed at the Institute for Parasitology, University of Veterinary Medicine Hannover, Foundation, Hannover, Germany. This included faecal ingesta samples analysed with the combined sedimentation–flotation and McMaster method, and faecal samples from hares not previously frozen subjected to the Baermann migration method. The methods are described in detail in a previous study [43].

Of note, *Trichostrongylus* spp. and *Graphidium* (*G*.) *strigosum (Dujardin, 1845)* eggs were not differentiated in 2016 but summarised as gastrointestinal nematodes. Accordingly, statistical analyses regarding distribution of these two parasites did not include the data of 2016.

### 2.4. Microbiology

Standard swab samples (TS-Swab Pl-Amies, sterile; HEINZ HERENZ Medizinalbedarf GmbH, Hamburg, Germany) from the intestine and the lung were taken for microbiological analyses. Additional swabs or tissues were sampled in cases of macroscopic organ alterations. Swabs were stored at 4 °C and tissue samples at −20 °C. Samples were cultivated in 2016 and 2017 at the Institute for Microbiology, University of Veterinary Medicine Hannover, Foundation, Hannover, Germany and in 2019 and 2020 at the Institute of Hygiene and Infectious Diseases of animals, Justus Liebig University Giessen, Giessen, Germany.

In the first two years of the study, intestinal samples were cultured on Columbia agar with sheep blood (Oxoid, Wesel, Germany) and Gassner agar (VWR, Darmstadt, Germany). The swab was then washed out in two Salmonella-selective enrichment broths, tetrathionate brilliant green bile broth (TBG; VWR), and Rappaport Vassiliadis Soy broth (RVS; Oxoid). Media were incubated at 37 °C overnight. On the following day, the liquid media were streaked on solid Brilliance™ Salmonella agar (Oxoid). Cultures were screened for *Escherichia (E.) coli* (*Migula, 1895*), *Aeromonas* spp., and *Salmonella* spp. after 24 h and 48 h incubation. Lung samples were streaked on Columbia agar with sheep blood (Oxoid), Gassner agar (VWR), Staph/Strep selective medium (in house), *Pasteurella* selective agar with Neomycin, and Bacitracin (inhouse) and placed into nutrient enrichment broth (inhouse). Plates and nutrient broth were incubated overnight at 37 °C; afterwards, nutrient broth was streaked out on the same media as mentioned above for intestinal samples and incubated again 24 h at 37 °C. Cultures were screened for potential pathogens including *P. multocida* and *Bordetella* (*B*.) *bronchiseptica* (*Ferry, 1912*). Isolates were identified using MALDI-TOF mass spectrometry (Bruker Daltonics, Bremen, Germany) and, if necessary, using additional biochemical tests.

From 2019, microbiological samples were processed further, as described by Faehndrich et al. [43].

In all study years, hares were tested for *F*. *tularensis* at the Institute of Bacterial Infections and Zoonoses, Federal Research Institute for Animal Health, Jena, Germany, as described in Faehndrich et al. [43]. For three hares suspected to suffer from brucellosis, additional serological and molecular detection methods for *Brucella* spp. were conducted in addition to cultivation. Serum samples of each animal were analysed using the Rose Bengal antigen reaction, serum agglutination test and complement fixation test [72]. Furthermore, PCR assays were conducted on DNA, extracted from liver tissue, to type the isolates to biovar level [72].

### 2.5. Virology and Serology

At the Istituto Zooprofilattico Sperimentale della Lombardia e dell’Emilia Romagna (ISZLER), Brescia, Italy, liver tissue samples and serum or “liver juice” samples were screened for EBHSV and RHDV2. Virological and serological analyses as well as interpretations were conducted, as described in Faehndrich et al. [43].

### 2.6. Statistical Analyses

The pathomorphological findings were our focus of interest for comparing the hunting grounds. Secondary analyses were conducted to clarify the aetiology of these pathomorphological findings.

Statistical analyses were performed only for pathomorphological findings, which were observed frequently in this study, and which might have an infectious origin. We assessed the effects of age, sex, year of sampling, and origin of sample on the prevalence of each of the selected pathomorphological findings using logistic regression. For each pathomorphological finding, the full model contained the main effect predictors (age, sex, year of sampling, and origin of sample) and all interactions up to the four-way interaction between all main effect predictors. The year of sampling was coded as a factor variable to allow for fluctuations in prevalence between years. Model selection was based on the small sample corrected Akaike information criterion (AICc) and an exhaustive screening of all nested models using function dredge in the R-package MuMIn. The significance of the parameters in the highest-ranking model was assessed based on likelihood ratio tests that compare the likelihood of the final model to the likelihood of the model in which the focal parameter was dropped. Post-hoc comparisons were applied for predictors with a significant effect on the prevalence of pathomorphological findings (Appendix A). To control for multiple testing, *p*-values in post-hoc comparisons were adjusted using the Holm–Bonferroni correction.

For evaluating signalment and biometric, parasitological, microbiological, and virological data, we used two-sample tests for the equality of proportions to test for the difference between two proportions and to assess whether there were statistically significant differences by using the function “prop.test” in the R-package stats.

All statistical analyses were performed in R, v 4.2.2 (R Foundation for Statistical Computing).

## 3. Results

Analysed hares include 78 females and 77 males (Table 1). Most of the sampled hares were classified as adults (65.2%, n = 101), according to eye lens weight. Sample composition with respect to age structure and sex ratio differed between the three hunting grounds and survey years (Table 1). Adults outnumbered juveniles in all three hunting grounds (Table 1). This was significant for Tetenbüll (X^2^ (1, N = 52) = 6.9423, *p* = 0.008) and Friedrichskoog (X^2^ (1, N = 55) = 7.2727, *p* = 0.007), but not for Elpersbüttel (X^2^ (1, N = 48) = 0.52083, *p* = 0.47). 

The sex ratio was slightly biased towards males in Tetenbüll, towards females in Elpersbüttel, and unbiased in Friedrichskoog, without any difference being significant. For the distribution of age categories over the sampling years, every year, more adults than juveniles were sampled, except for the year 2019, which revealed the highest proportion of juveniles (63.0%, n *=* 17, X^2^ (3, N = 155) = 13.371, *p* = 0.004). Only in Tetenbüll´s sampled hares from 2019, adults still outnumbered juveniles, although even here, the highest proportion of sampled juveniles was detected (J/A = 0.80).

The average body weight of sampled hares was 3743 g with a minimum of 1589 g (juvenile, male, Tetenbüll, 2017) and a maximum of 4953 g (adult, female, Friedrichskoog, 2017). There were almost no differences in body weight between the sexes (average body weight males: 3686 g; average body weight females: 3799 g), whereas clear differences were observed between juveniles (average body weight: 3376 g (average age: 174 days)) and adults (average body weight: 3939 g). Compared to Elpersbüttel (3796 g) and Friedrichskoog (3845 g), hares from Tetenbüll revealed the lowest average body weight (3585 g). This was also the case when differentiating adults (Tetenbüll: 3809 g; Friedrichskoog: 4014 g; Elpersbüttel: 4005 g) from juveniles (Tetenbüll: 3082 g; Friedrichskoog: 3467 g; Elpersbüttel: 3527 g), although juveniles were almost of the same average age (Tetenbüll: 175 days; Friedrichskoog: 171 days; Elpersbüttel: 176 days).

The body condition of most animals was classified as good (71.0%, n = 110). One-quarter of the animals was in a moderate (24.5%, n = 38) and just a few animals in a poor body condition (4.5%, n = 7). A poor body condition was only observed in hares from Tetenbüll (13.5%, n = 7, X^2^ (2, N = 155) = 14.521, *p* = 0.0007). For sampling years and sex, no significant difference was detected.

### 3.1. Pathomorphological Findings

Preservation status and hunting-induced lesions permitted the assessment of routinely taken samples of the lung (n = 155), heart (n = 155), liver (n = 155), spleen (n = 155), kidney (n = 155), skeletal muscle (n = 154), retroperitoneal fat (n = 155), brain (n = 148), mesenteric lymph nodes (n = 155), small intestine (n = 155), large intestine (n = 155), and adrenal glands (n = 146) for histopathology.

Several pathomorphological findings within various organ systems were detected in histopathological analyses (Appendix A). Overall, the alimentary system was the most affected body system, with thirteen different pathomorphological findings, where at least one of these was present in 91.0% (n = 141) of examined individuals. The next frequently affected body systems were the haematopoietic and endocrine systems with eleven different pathomorphological findings (57.4%, n = 89), the urinary and genital system with five (31.6%, n = 49), the abdominal and thoracic cavities with eight (23.2%, n = 36), and the respiratory system with four different pathomorphological findings (9.7%, n = 15). Only two pathomorphological findings were present in the cardiovascular system in 1.3% (n = 2) of hunted hares and one pathology in the skin and nervous system in 0.6% (n = 1) of hares, respectively. No relevant pathomorphological findings were detected in the locomotory system. Overall, during histopathological analyses, detected intestinal parasitism (63.9%, n = 99) and hepatitis (55.5%, n = 86) were the most frequently recorded pathomorphological findings in hares in the investigated hunting grounds. In four hares, no pathomorphological findings were detected other than hunting-related and agonal ones. However, hunting-related tissue and organ destructions may have influenced evaluation of pathomorphological findings.

Hereinafter, we focus our analysis on pathomorphological findings, which were observed frequently in this study, and which might have an infectious origin (Table 2): hepatitis, enteritis, steatitis, pneumonia, nephritis, and, during histopathological analyses, detected parasites in the intestine. Other detected pathomorphological findings are mentioned only in association with these (Appendix A).

#### 3.1.1. Alimentary System

**Liver:** Among the six pathomorphological findings analysed in depth, hepatitis was the second most common and was observed in 55.5% (n = 86) of the studied individuals. Sampling year (*p* = 0.0002; Appendix A) and age (*p* = 0.002; Appendix A) proved to have a significant effect on the presence of hepatitis whereas hunting ground and sex did not reveal a significant effect. Highest levels of hepatitis were observed in 2016 (Figure 3a). Post-hoc analyses confirmed that the drop to lower levels in 2017 (*p* = 0.0003; Appendix A) and 2020 (*p* = 0.041; Appendix A) was significant. Adults were significantly more frequently affected than juveniles (*p* = 0.002; Appendix A; Figure 3b).

The main type of hepatitis was non-purulent with different proportions of lymphocytes, plasma cells, and macrophages (93.0%, n = 80; Figure 4), followed by purulent (14.0%, n = 12), necrotising (5.8%, n = 5), and granulomatous type (5.8%, n = 5). Eight cases (9.3%) of hepatitis showed a mixed inflammatory type.

Occasionally, hepatic fibrosis (15.1%, n = 13) or liver cell necrosis (15.1%, n = 13) were associated with hepatitis (Appendix A). In one adult female hare from Tetenbüll, mixed non-purulent to purulent hepatitis was associated with purulent cholangitis and bile duct proliferation, and in seven others (Tetenbüll: n = 4; Friedrichskoog: n = 3), only bile duct proliferation was present next to hepatitis. Granulomatous hepatitis with liver cell necrosis and dystrophic calcification of an adult male from Friedrichskoog was associated with cestode cysts on the liver surface. Another adult female from Friedrichskoog showed granulomatous to necrotising lymphadenitis of mesenteric lymph nodes with simultaneously occurring granulomatous to necrotising hepatitis.

**Intestine:** The most common finding in histopathological analyses, which was observed in 99 out of 155 investigated hares (63.9%), was the detection of parasites or parasite stages in the intestine. Mainly different stages of protozoal organisms were found (74.7%, n = 74) and either detected in the intestinal epithelium (n = 35, Figure 5) or in the intestinal lumen (n = 19).

In two animals, a cestode or its metacestode (2.0%), and in 30 animals, nematodes (30.3%) were found in the intestine during pathomorphological investigations. All these parasitological findings during pathomorphological analyses correspond with the results of parasitological analyses.

The presence of parasites or parasite stages in the intestine was significantly affected by the interaction of sex and age (*p* = 0.048; Appendix A). Parasites were detected significantly more frequently in male juveniles than in male adults (post-hoc test, *p* = 0.038; Appendix A; Figure 6).

In 20 animals (20.2%), enteritis and parasites or parasite stages were detected simultaneously in histopathological investigations.

An activation of the regional immune system in the form of hyperplasia of mesenteric lymph nodes or Peyer’s patches was diagnosed concurrently with intestinal parasites in 47 hares (47.5%).

Enteritis was one of the less frequent findings (13.5%, n = 21) with regard to the six explicitly analysed pathomorphological findings. The occurrence of enteritis was not significantly affected by sex and age category. Regarding the hunting grounds, enteritis was diagnosed the most in hares sampled in Tetenbüll (21.2%, n = 11), followed by those from Elpersbüttel (12.5%, n = 6) and Friedrichskoog (7.3%, n = 4). The predictor hunting ground was retained in the model selection process based on AICc, and a *p*-value of 0.097 was obtained in likelihood ratio tests, so that we can neither rule out nor confirm an effect of hunting ground on the prevalence of enteritis (Figure 7a). On the other hand, a significant effect was observed between sampling years (*p* = 0.027; Appendix A). Whereas inflammation of the intestine was not observed in 2019, at least 15% of the sampled individuals suffered from enteritis in the remaining sampling years (2016: 15.1%, n = 8; 2017: 18.8%, n = 9; 2020: 14.8%, n = 4) (Figure 7b).

In 81.0% of cases, the enteritis was non-purulent (n = 17) with various infiltrations of lymphocytes, plasma cells, histiocytes, and partly eosinophils (Figure 8), and in 33.3%, a granulomatous (n = 7) inflammatory type was present. A mixed inflammatory type was detected in 14.3% (n = 3) of affected animals.

In some cases of enteritis, hyperplasia of the mesenteric lymph nodes (47.6%, n = 10) or Peyer’s patches (9.5%, n = 2) and thus an activation of the regional immune system was also present. In 50 hares (32.3%), unformed rectal faeces were detected macroscopically during necropsy, including 21 (43.8%) animals from Elpersbüttel, 15 (28.8%) from Tetenbüll, and 14 (25.5%) from Friedrichskoog. In 16 (32.0%) of these, hyperplasia of regional lymphatic tissue was diagnosed histopathologically (mesenteric lymph node: n = 14; Peyer´s patches: n = 1; both: n = 1) without simultaneously occurring non-catarrhal enteritis. Nine animals (Elpersbüttel: n = 6; Friedrichskoog: n = 1; Tetenbüll: n = 3) with enteritis were diagnosed additionally with unformed rectal faeces.

#### 3.1.2. Abdominal and Thoracic Cavities

**Adipose tissue:** The major finding in the retroperitoneal fat tissue was steatitis (23.2%, n = 36). Neither age, sex, hunting ground, nor sampling year have a significant effect on the presence of steatitis.

In 88.9% (n = 32) of cases, steatitis was granulomatous, including 31 cases (86.1%) of granulomatous to necrotising steatitis (Figure 9).

Two females, one juvenile and one adult, with mixed type steatitis also had dystrophic calcification of necrotic adipose tissue. One adult male from Friedrichskoog with focal granulomatous steatitis had an associated cystic cestode infection. In another hare (adult, female, Elpersbüttel) with granulomatous to necrotising steatitis, simultaneously, lymphadenitis of mesenteric lymph nodes with the same inflammatory type was identified.

Five cases showed infiltrations of lymphocytes, macrophages, and plasma cells. In eight affected animals next to routinely sampled retroperitoneal fat tissue, other fat depots of the body (pericardial and mesenterial) also showed steatitis histopathologically. Parallel to non-purulent steatitis, a mixed type pyogranulomatous and partly non-purulent serositis of the large intestine was present in a juvenile male from Tetenbüll.

#### 3.1.3. Respiratory System

**Lung:** In eight cases (5.2%), the hunted hares exhibited pneumonia. The prevalence in females (9.0%, n = 7) was significantly higher compared to males (1.3%, n = 1, *p* = 0.022; Figure 10). Age, hunting ground, and sampling year had no significant effect on prevalence of pneumonia.

Pneumonia cases were mainly non-purulent with infiltrates of lymphocytes, histiocytes and plasma cells (75.0%, n = 6). One hare additionally showed a chronic focal pleuritis associated with cestode infection. The remaining two cases were either purulent or granulomatous (12.5%, n = 1, respectively). The one hare with purulent pneumonia, revealed additionally hyperplasia of the bronchus-associated lymphoid tissue.

#### 3.1.4. Urinary and Genital System

**Kidneys:** In one-third of the studied animals, inflammation of the kidneys was found (31.0%, n = 48). Whereas no significant effect of hunting ground or sex was observed, age (*p* = 0.01, Appendix A) and sampling year (*p* = 0.007, Appendix A) revealed a significant effect on the presence of nephritis. The prevalence of nephritis was the highest in 2016 (47.2%, n = 25; Figure 11a), with significantly more hares confirmed in post-hoc analyses compared to hares sampled in 2017 (18.8%, n = 9, *p* = 0.009; Appendix A). Between the other sampling years, no significant difference was determined in statistical analyses. Furthermore, significantly more adults were diagnosed with nephritis (37.6%, n = 38) than juveniles (18.5%, n = 10, *p* = 0.01; Figure 11b).

All inflammations of the kidneys were non-purulent, with interstitial infiltrations of lymphocytes, macrophages, and plasma cells (Figure 12). In one adult male, fibrosis and depression on the kidney surface were observed.

### 3.2. Parasitology

On 147 hares, McMaster and combined sedimentation–flotation were conducted, while only one method was used on four hares, respectively. Furthermore, faecal samples of 54 hares were subjected to the Baermann migration method. Except for detected cestodes, all parasitological results presented hereafter refer to parasite eggs or oocysts.

No parasitic stages were detected in three adult males (1.9%) from 2016, one from each hunting ground. In two of these (adult males from Tetenbüll and Elpersbüttel), oocysts and nematodes, respectively, were detected in the intestinal lumen during histopathological examination. *Eimeria* spp. were identified in 149 animals (96.1%), and 74 of these samples yielded enough faecal material for in vitro sporulation to identify the organisms at species level (Table 3). As in almost all hares, *Eimeria* spp. was isolated, and analyses on differences in hunting ground, sampling year, age category or sex were not meaningful.

An endoparasitic coinfection was present in 112 hares, and in 110 of these individuals *Eimeria* spp. was identified. A coinfection was least common in sampling year 2016 (45.3%, n = 24, X^2^ (3, N = 112) = 30.47, *p* = 0.000001) compared with the other years (>80%).

*Trichostrongylus* spp. was detected in 53 hares (52.0%), each diagnosed together with *Eimeria* spp., 10 additionally with *Trichuris* spp., and 11 additionally with *G. strigosum*. The prevalence of *Trichostrongylus* spp. was significantly higher in 2017 (81.3%, n = 39, X^2^ (2, N = 102) = 35.908, *p* = 0.00000002) and the intensity of excretion was between 0 and 1067 eggs per gram faeces (epg) in this sampling year. Most of this parasite species were detected in hares from Friedrichskoog (65.7%, n = 23), followed by those from Tetenbüll (54.3%, n = 19) and Elpersbüttel (34.4%, n = 11). This difference in hunting grounds proved to be significant (X^2^ (2, N = 102) = 6.6927, *p* = 0.035).

Other detected endoparasites were *G. strigosum* (41.2%, n = 42), *Trichuris* spp. (15.9%, n = 24), *Passalurus* spp. (4.0%, n = 6), *Capillaria* spp. (5.1%, n = 5), *Strongyloides* spp. (1.3%, n = 2), and the cestodes *Mosgovoyia pectinata (Goeze, 1782)* (adult worm; n = 1) and *Taenia* (*T*.) *pisiformis (Bloch, 1780)* (metacestode; n = 1). The latter parasite was diagnosed in a hare showing cystic cestode infection at serous membranes in the pathomorphological examination (Figure 13).

Gastrointestinal nematodes were detected in 19 hares (35.8%) in 2016. *G. strigosum* was not discovered in 2017, but, on the contrary, in more than 70% of hares tested in 2019 (n = 22) and 2020 (n = 20), which represented a significant difference (X^2^ (2, N = 102) = 63.772, *p* < 0.000001). The intensity of excreted eggs of *G. strigosum* ranged in 2019 between 0 and 1133 epg and in 2020 between 0 and 400 epg. Of those infected by *Trichuris* spp., 19 animals were sampled from the hunting ground Friedrichskoog, constituting more than one-third of sampled animals from this location (36.5%). The other hunting grounds revealed less than 10% *Trichuris* spp. identifications (Tetenbüll: 7.7%, n = 4; Elpersbüttel: 2.1%, n = 1) and this difference was statistically significant (X^2^ (2, N = 151) = 25.858, *p* = 0.000002). No lungworms, either at larval or adult stages, were detected in any of the samples examined.

For seven individuals, consisting of significantly more juveniles (11.1%, n = 6, *p* = 0.008), a relevant excretion of coccidia oocysts (>100,000 opg) was detected with the McMaster method. No significant differences in relevant excretion were found for hunting grounds, sampling year, and sex. The average weight of affected juveniles was 2423 g at an average age of 124 days of life, while not affected juveniles in our study of the same age weighted more (average: 3057 g, 119 days of life). On the other hand, the only affected adult female revealed a weight above the average for adult hares in our study (4214 g, average: 3939 g). Only one affected juvenile female showed a poor body condition (body weight: 2371 g; age: 188 days), whereas all other affected animals were good nurtured. In five of these hares with relevant coccidia oocysts excretion, hyperplasia of mesenteric lymph node was diagnosed in pathomorphological examination, including two animals with unformed rectal content.

For one hare showing cystic cestode infection at serous membrane of the liver in the pathomorphological examination, metacestodes of *T. pisiformis* were identified in parasitological analyses (Figure 13).

In almost every case of enteritis, *Eimeria* spp. (n = 19) was identified, followed by *Trichostrongylus* spp. (n = 8), *Trichuris* spp. (n = 2), *Passalurus* spp. (n = 2), and *Capillaria* spp. (n = 1). In 90% of enteritis cases, coinfection of at least two parasite species could be detected.

None of the hares tested positive for *E. stiedai* were diagnosed with bile duct alterations in the pathomorphological examination, but 35% (n = 6) had non-purulent (n = 4), pyogranulomatous (n = 1), or purulent to non-purulent (n = 1) hepatitis. Two adult females from Tetenbüll and Friedrichskoog with proof of *G. strigosum* showed non-purulent gastritis in histopathology.

### 3.3. Microbiology

Microbiological samples were taken from various organs, including kidneys, liver, reproductive tract, ear, trachea, mesenteric lymph node, and spleen, but primarily from intestine and lung. All hares were sampled, and 81 different species or genera of bacteria and fungi were detected (Appendix A). *E*. *coli* was by far the most common cultured organism (n = 276, 145 animals) and was detected in all sampled organ systems, followed by *Pseudomonas* (*P*.) spp. (n = 133, 99 animals) and *Aeromonas* (*A*.) spp. (n = 73, 59 animals). Hereinafter, only the bacteria and fungi, previously described as pathogens in *L*. *europaeus*, are discussed. Pathogenic bacteria detected were *Brucella* (*B*.) *suis (Huddleson, 1929)* Biovar 2 (n = 2), *B*. *bronchiseptica* (n = 1), and *Yersinia* (*Y*.) *enterocolitica (Schleifstein and Coleman, 1939)* (n = 1). The two *B*. *suis* Biovar 2 isolates were discovered in the livers of one adult female (Figure 14) and one adult male from the hunting ground Elpersbüttel sampled in 2017. The adult male was diagnosed with mixed non-purulent to granulomatous hepatitis, pyogranulomatous enteritis and granulomatous to necrotising orchitis. The adult female did show non-purulent hepatitis. 

The isolations of *B*. *bronchiseptica* from the lung of an adult male from Tetenbüll, sampled in 2016, and the detection of *Y*. *enterocolitica* in the intestine of a juvenile male from Friedrichskoog, sampled in 2019, were also not associated with typical pathomorphological findings in the respective organs.

One male and one female adult from Friedrichskoog sampled in 2016 as well as one female adult from Tetenbüll sampled in 2017 revealed the presence of *Klebsiella* spp. in the lung but were not associated with any pathomorphological findings.

### 3.4. Virology and Serology

Almost half (n = 67) of the sampled *L*. *europaeus* were tested for a current or overcome lagovirus infection detected using either virological or serological testing or a combination of both.

All samples were screened using serology, but one hare was only tested for lagovirus, not including specific analysis on certain species. For 66 animals, serology was conducted for RHDV2 and EBHSV, including 53 serum and 13 liver juice samples.

The overall seroprevalence for EBHSV was 90.9% (n = 60), including those with an RT value (ratio between titres against EBHSV to those of RHDV2) equal to 4, which is the lower limit to define that antibodies are specific against one virus. One tested hare (1.4%; juvenile, male, Elpersbüttel) from 2019 revealed a serotitre with a RT = 0.25 and a titre of 40, indicating an old, overcome infection with RHDV2. For six hares (9.0%), the exact determination of the virus that induced antibodies was impossible. None of the tested hares were seronegative for lagovirus.

According to the titre in cELISA, 52 hares were conclusive for an overcome EBHSV infection (78.8%, RT value > 4), including 15 animals being certainly infected within a few months (EBHSV titres ≥ 640), as well as one naïve hare (IgM anti-EBHSV ≥ 1280–2560) infected within a few weeks before sampling (EBHSV titres ≥ 2560). For eight hares, an overcome EBHSV infection was suspected (12.1%, RT value = 4). There was no indication of a current infection at point of sampling in any of the tested hares.

No significant difference in occurrence of recent to very recent infection was detected for age, hunting ground, sampling year, or sex. Slight differences in EBHSV median serotitres were detected between age categories (juveniles: 320; adults: 160), hunting grounds (Tetenbüll: 320; Friedrichskoog: 320; Elpersbüttel: 160), sex (males: 160; females: 320), and sampling years (2016: 320; 2019: 320; 2020: 160), but were considered not significant.

Typical pathomorphological alterations such as hepatocellular necrosis were diagnosed in seven (10.4%) lagovirus seropositive animals, mainly including animals sampled from Friedrichskoog (n = 6) and adults (n = 5). Six mainly adult animals (n = 5) with pathognomonic hepatocellular necrosis could not be tested for lagovirus infection. Furthermore, hepatitis was correlated with certain EBHSV infections (n = 29), uncertain EBHSV or RHDV2 infections (n = 4), and not further determinable lagovirus infections (n = 2). Following previous studies on EBHSV [30,49], a suspected chronic or subclinical form with mild lymphocytic hepatitis was detected in 53.3% (n = 32) of EBHSV-seropositive cases (RT ≥ 4), including nine hares with recent to very recent infection.

This chronic, lymphocytic hepatitis with concurrent positive EBHSV serotitre (RT ≥ 4) was only significantly associated with age (X^2^ (1, N = 60) = 4.0191, *p* = 0.045), with more adults being affected (64.9%, n = 24) than juveniles (34.8%, n = 8).

## 4. Discussion

Scientists and hunting associations in different parts of Europe are concerned about *L*. *europaeus*, as population sizes have been decreasing for several years [3,4,5,25]. However, in some areas, population sizes are developing widely differently, as also observed for hares of the northernmost German federal state Schleswig-Holstein [14]. While the current status of infectious diseases in *L*. *europaeus* has been assessed in a nationwide study within Schleswig-Holstein through examination of dead hares from passive surveillance [43], this study focusses on the health situation in regional hare populations. In detail, we aimed to assess if population variables are associated with pathomorphological findings and infectious diseases in order to gain insight into the diverse dynamics of different brown hare populations in Schleswig-Holstein. Therefore, we evaluated the prevalence of selected pathomorphological findings together with infectious pathogens and their association with location, year, sex, and age of animals in 155 hares from three hunting grounds with high-density populations between 2016 and 2020 in Schleswig-Holstein.

Overall, our results indicate a low prevalence of clinically relevant pathogens in the investigated populations. Sampling hunted animals, which is supposed to be for human consumption, is likely biased towards rather more healthy specimens and only provides insights into the situation during hunting season. However, hunted animals are appropriate for collecting data on demographic population structures as well as the prevalence of infectious diseases within populations and can be standardised more easily for longitudinal follow-up studies. For comprehensive evaluation of the population’s health status, this should be complemented with investigations on deceased animals to collect data on different causes of death to draw conclusions concerning the impact of pathogens on the population level. After performing a parallel health assessment of hunted and deceased hares in the same area [43], we have learned that only the combination of both strategies is able to provide a realistic insight into the health status of the population as both investigations individually have major limitations.

The results of this study indicate some significant associations of location, year, age, and sex of animals and selected pathomorphological findings as well as infectious pathogens in sampled hare populations of Schleswig-Holstein.

In general, sample composition regarding sex and age ratio was in agreement with previous data of hunted hares from this federal state [44,45] and showed a comparable development for all three hunting grounds over the study years. Adults exceeded juveniles in almost all years and hunting grounds. According to Zörner [71], this age ratio at the end of the year and reproductive period is characteristic for an environment with a decreased survival rate of juveniles. An exception to this were the hunting grounds Elpersbüttel and Friedrichskoog in the sampling year 2019, with proportionally more juveniles. An increase in juveniles without a reversion of age composition was also seen in the hunting ground Tetenbüll. In Elpersbüttel, the age composition was more balanced compared to the other sampled hunting grounds. The reasons for differences in age distributions can be various, but juvenile mortality in particular is a parameter under discussion [73,74,75,76,77]. In another study, undertaken year-round, parallel to this one, on deceased hares, juveniles exceeded adults [43]. This indicates a high juvenile mortality rate in the time before hunting takes place and might explain why we sampled more adults in this study on hunted hares.

Although sampled animals were generally in good body condition and average body weights were comparable to previous morphometric data [39,40,71], the hares with poor body condition were clustered in Tetenbüll. In this hunting ground, adults and juveniles had the lowest average weights from all hunting grounds, even though sampled juveniles had a comparable average age in all three locations. However, Tetenbüll was sampled each year some weeks before the other two hunting grounds. Still, as the average age of all hunting grounds was comparable, juveniles from Tetenbüll must have been born earlier in the year. One reason for the higher number of hares in poor body condition in Tetenbüll might be the regional habitat, with mainly grassland and less cultivation, probably offering less convenient food. In previous studies, areas with primarily arable land, especially with root crop cultivation, showed higher hare densities and were valued as better hare habitats than extensively managed grassland [40,41,78]. Another reason might be the high prevalence of enteritis in hares from Tetenbüll, with almost one-quarter of sampled hares affected. A definite cause for the enteritis cases was not identified. Interestingly, this pathomorphological finding was not diagnosed in 2019, while in the same year, the proportion of juveniles increased. A simultaneously conducted study on deceased hares in this federal state also reported less cases of enteritis in 2019 compared to other study years [43]. Particularly due to less precipitation and higher temperatures in some years, the usually high parasitic pressure can be decreased [71,79], and thus possibly also the prevalence of enteritis, which can be caused by high infection levels of parasites [61,62,66,80]. In 2019, the summer was very dry and hot [81]. However, when merging pathomorphological and parasitological data, almost every hare examined in our study was proven to excrete parasite eggs or oocysts (96.1%). A high intensity of parasitic infections in *L*. *europaeus*, especially with coccidia, is already described [39,56,82], but healthy adults are known to tolerate a certain level of parasite infection without negative effects on individual health [56]. Here, we could confirm the results of previous studies [46,47,80,83] referring to juveniles being more often infected with parasites and revealing significantly higher excretion rates more often than adults. Especially in subadult hares, enteritis with concurrent diarrhoea caused by severe coccidiosis can lead to emaciation of animals [66,71]. This might be explained by the naïve immune system of juveniles, resulting in higher infection prevalence and oocyst excretion rates. In the present study, however, significantly higher frequencies of coccidiosis were only observed for juvenile males but not for females. One explanation might be a limited validity of this histopathologically generated parasitological data, as only limited areas of the intestine could be examined. Previous studies in other species describe an effect of hormones on susceptibility to parasitic infections, with androgens increasing infection rate with parasites and inhibiting antibody formation [84,85,86]. In other parts of Germany, females were infected as often as males with *Eimeria* spp. but revealed a higher excretion rate [56]. Another significant local difference in parasitic infection is the higher prevalence of *Trichostrongylus* spp. and *Trichuris* spp. in Friedrichskoog compared to the other hunting grounds. Especially in *T*. *retortaeformis*, a common gastrointestinal nematode in *L*. *europaeus*, hypobiosis is a common feature in times of challenging environmental conditions [87]. This might be the reason for various excretion rates over the years, and determined prevalence may even be underestimated, as sampling was conducted in early winter when hypobiosis mainly occurs in central Europe. Furthermore, the detection of this parasite is interesting as previous research on mountain hares (*Lepus* (*L*.) *timidus*) showed increased fecundity in females by decreasing the abundance of *Trichostrongylus* spp. [88]. It was suggested that this increased fecundity might be associated to better body condition in these female *L*. *timidus* following reduction of parasite burden [89,90]. An influence of *T*. *retortaeformis* on body weight was also confirmed in a study conducted in Austria [80]. In our study, poor body condition was only detected in Tetenbüll, where *Trichostrongylus* spp. was diagnosed frequently. Consequently, high prevalence of *Trichostrongylus* spp. should be considered as an impact factor on population development in this region. However, it is important to mention that despite high prevalence of intestinal parasites, enteritis was not a very common finding, which, on the other hand, indicates a rather low importance for the health status of the studied populations. This corresponds with another study, conducted in this region and which ascribed parasites a low relevance for population decline [41].

Regarding the pathomorphological finding pneumonia, a difference between sexes was detected with more females affected. Common infectious pathogens within *L*. *europaeus* causing pneumonia are lungworms or *Toxoplasma gondii* (*Nicolle and Manceaux, 1908*), but the first was not detected during this study, and the latter was not tested out specifically. Non-infectious causes should be considered as well but were out of the scope of this study. So, we could neither identify the aetiology of pneumonia nor the reason for the higher prevalence in females. However, with a comparably low prevalence in hunted hares during this study and even in deceased hares sampled in the same federal state [43], this predominantly non-purulent type of pneumonia is of rather minor relevance for the health status of *L. europaeus* in this region.

The pathomorphological findings of hepatitis and nephritis appeared to be age-related and were significantly more common in adults than in juveniles. Both findings were mainly diagnosed as chronic, non-purulent alterations and were already detected in deceased hares in this area with lower prevalence [43] and again more often in adults than in juveniles. Previous studies correlated chronic, especially lymphocytic hepatitis to chronic infection with the lagovirus EBHSV [30,49]. In our study, lymphocytic hepatitis together with EBHSV titre was detected in only half of the tested hares. However, the combination of chronic, lymphocytic hepatitis and EBHSV-seropositive titres was significantly more often detected in adults. Moreover, although not significant, adults were more often diagnosed with lower EBHSV titres compared to juveniles, indicating a decreased specific immune response towards EBHSV in adults. This supports the assumption that diagnosed chronic hepatitis might have been caused by EBHSV. However, no intralesional virus detection, such as immunohistochemistry, was performed to prove this hypothesis and so other infectious and non-infectious noxae could not be ruled out. With 90.9%, the seroprevalence for EBHSV in this study was higher as reported in previous studies on hunted [32,35,36,45] and deceased hares of Schleswig-Holstein [43]. However, typical acute phase pathomorphological alterations such as hepatic necrosis were detected only in 10.4% of lagovirus-seropositive animals, indicating an endemic situation with a stable virus circulation in the tested populations.

By analysing the impact of the sampling year, again, the two pathomorphological findings hepatitis and nephritis were detected significantly more often in 2016 compared to 2017 and for hepatitis also compared to 2020. The comparison of median serotitres for EBHSV showed a higher serotitre for 2016 compared to 2020. This indicates a more recent immune response to EBHSV in 2016. This might also explain the higher prevalence of hepatitis for this year. For the sampling year 2017, we cannot conclude any correlation, as we did not test specifically for EBHSV. Nephritis was previously detected due to infections with *Encephalitozoon* spp. [31,54], which was not tested out in this study.

Interestingly, we identified a relatively high prevalence of hares with granulomatous-necrotising steatitis. This is in agreement with recent investigations on deceased hares in this area [43]. We hypothesised that this might be associated with a deficiency in food ingredients [91,92]. Thus, we expected to find local differences by sampling three different hunting grounds with at least two different types of agricultural utilisation for several years in this study. Unfortunately, we did not detect any significant differences in this pathomorphological finding. Future studies should therefore focus on the relationship between food and the prevalence of steatitis with in-depth analyses of diet components.

Except for occasional and localised proof of *Brucella* spp., no other detected potentially pathogenic bacteria were associated to disease in this study. Other studies suggested decreasing importance of bacterial pathogens for *L*. *europaeus* in Schleswig-Holstein as well [40]. At least a high seroprevalence and a wide distribution were reported recently in this federal state for *Yersinia* spp. [32], formerly known as one of the most important lethal bacteria for *L*. *europaeus*, with population losses of up to 50% [61]. Another recent study conducted on the island of Pellworm in Schleswig-Holstein, suggested an impact through the predominance of *Enterobacterales*, e.g., *E*. *coli*, in the intestinal microbiome on regional population decline [45]. Neither in the present study on hunted hares nor in our previous study on deceased hares in Schleswig-Holstein [43] were bacterial infections relevant to brown hare populations detected. Nevertheless, the prevalence of pathogenic bacteria was higher in deceased compared to hunted hares, as also reported in other studies [44,46]. This was expected, as it is more likely to detect pathogens in deceased animals and sampling period is not limited to hunting season. So, it might be that infected animals were already deceased before our sampling was conducted in early winter and so did not occur frequently in sampled hunted hares. Widespread serological studies can generate more information about the occurrence and relevance of pathogens in the population and should be considered for future studies.

The sampling of hunting bags was suitable for obtaining sufficient samples from all three hunting grounds across almost the whole study period. Only for 2018 could no data be collected, as no funding was available for that year. Excluding that year, consecutive, intermediate-term data could be generated for specific populations of *L*. *europaeus*.

A limitation of this study was the varying sample sizes across the years that had to be considered for data analysis in terms of comparability and validity of data. Sample sizes within the same years were comparable for all hunting grounds.

In conclusion, this study detected chronic inflammations conspicuously often, with significantly different prevalence, mainly in age categories and sampling years. Only a few infectious pathogens were present in the tested populations, and the aetiology of these frequent pathomorphological findings remains open. Overall, this indicates infectious diseases as of rather minor importance for observed population dynamics of *L*. *europaeus* populations in Schleswig-Holstein. Next to widespread serological analyses, future investigations should also consider non-infectious reasons such as intoxications and nutrient deficiencies to elucidate the cause of frequent occurring pathomorphological findings within the brown hare population of Schleswig-Holstein. 

## Figures and Tables

**Figure 1 pathogens-12-01317-f001:**
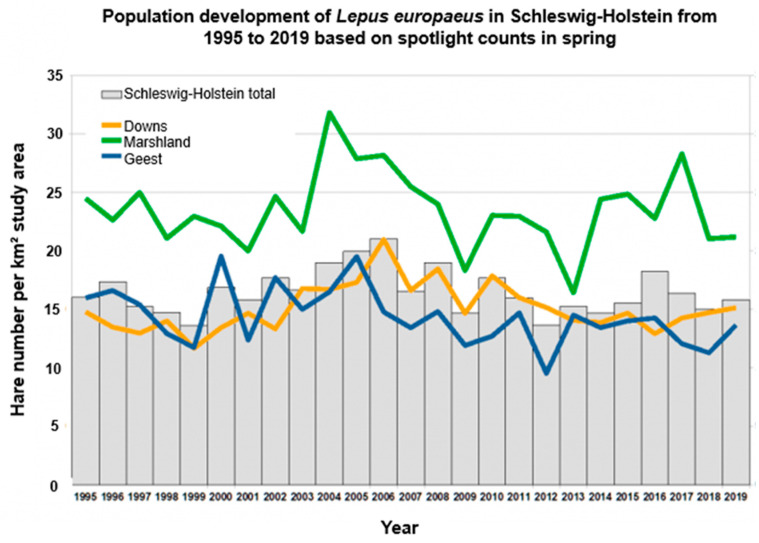
Development of population densities for *L. europaeus* in Schleswig-Holstein from 1995 to 2022 based on spotlight counts in spring with dynamic, but different population densities for whole Schleswig-Holstein and different natural regions (geest, downs, marshland) (adjusted from Wildtierkataster Schleswig-Holstein [12]).

**Figure 2 pathogens-12-01317-f002:**
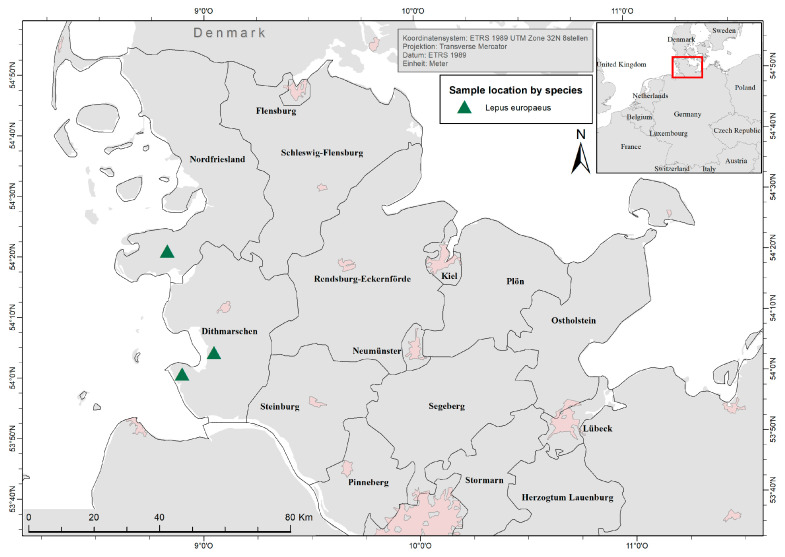
Locations of the three sampled hunting grounds (triangles). From north to south: Tetenbüll, Elpersbüttel, Friedrichskoog.

**Figure 3 pathogens-12-01317-f003:**
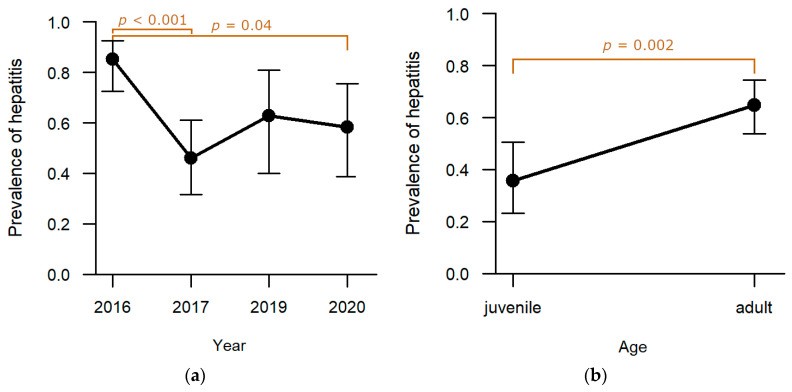
Effect of (**a**) sampling year and (**b**) age category on the prevalence of hepatitis. Confidence bars show the 95% confidence intervals around the estimates obtained through logistic regression. The effect of sampling year is depicted for adults (**a**), and the effect of age category is depicted for a weighted average across the sampling year effects with weights proportional to the number of samples obtained in each year (**b**). Significant differences between sampling years and age categories are illustrated individually using square brackets including the *p*-value and show a significant decrease in prevalence from 2017 to 2019 and 2020 as well as a higher prevalence of hepatitis in adults.

**Figure 4 pathogens-12-01317-f004:**
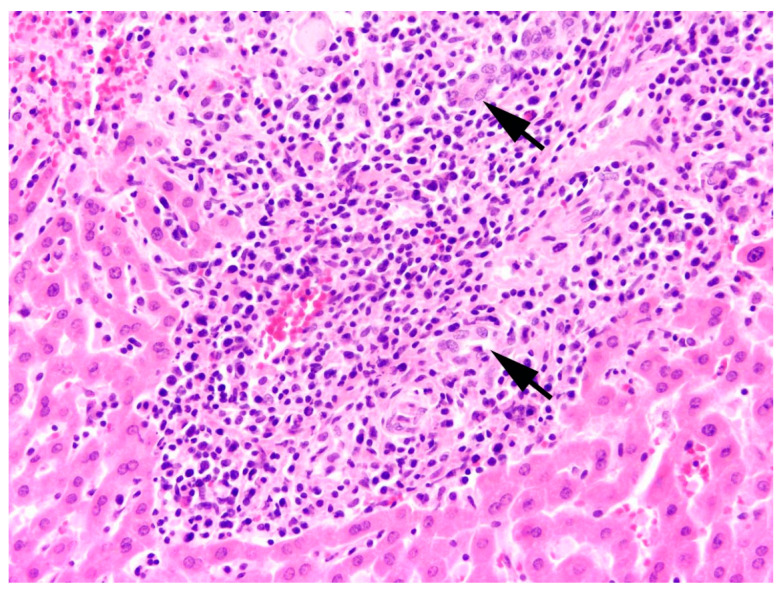
Liver, *L*. *europaeus*; severe infiltration of lymphocytes, macrophages, and plasma cells within a portal triad (arrows = bile ducts) and the periportal liver tissue. HE, 200×.

**Figure 5 pathogens-12-01317-f005:**
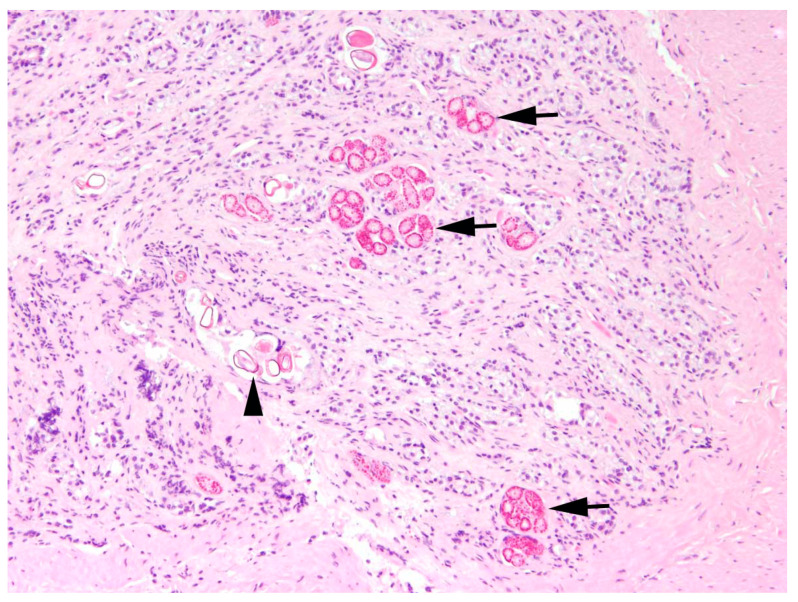
Intestine, *L*. *europaeus*; intestinal protozoonosis with coccidial stages in enterocytes (arrows) and oocysts in crypt lumina (arrowhead). HE, 100×.

**Figure 6 pathogens-12-01317-f006:**
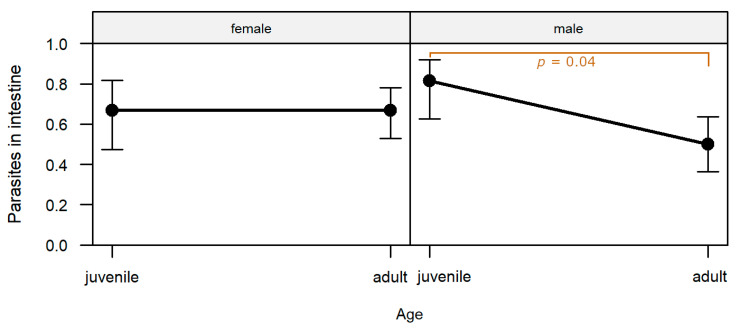
Effect of age and sex on the prevalence of parasites in the intestine. Confidence bars show the 95% confidence intervals around the estimates obtained through logistic regression. A significant difference was only detected within males between juveniles and adults and illustrated with a square bracket including the *p*-value.

**Figure 7 pathogens-12-01317-f007:**
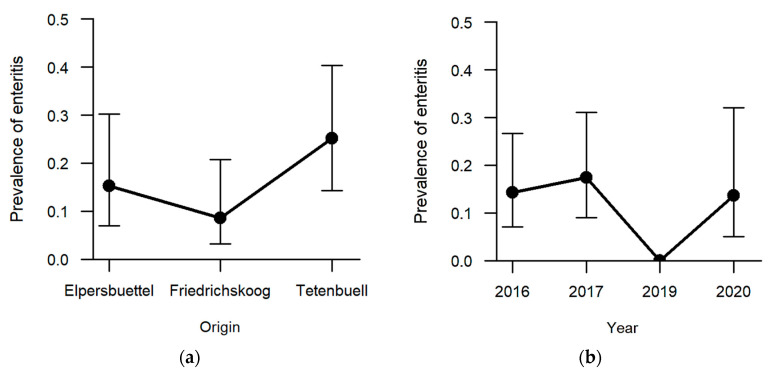
Effect of (**a**) hunting ground and (**b**) sampling year on the prevalence of enteritis. Confidence bars show the 95% confidence intervals around the estimates obtained through logistic regression. A significant effect of sampling years on prevalence of enteritis was detected in likelihood ratio tests, but post-hoc analyses did not reveal between which years this difference was present. In 2019, no enteritis was diagnosed.

**Figure 8 pathogens-12-01317-f008:**
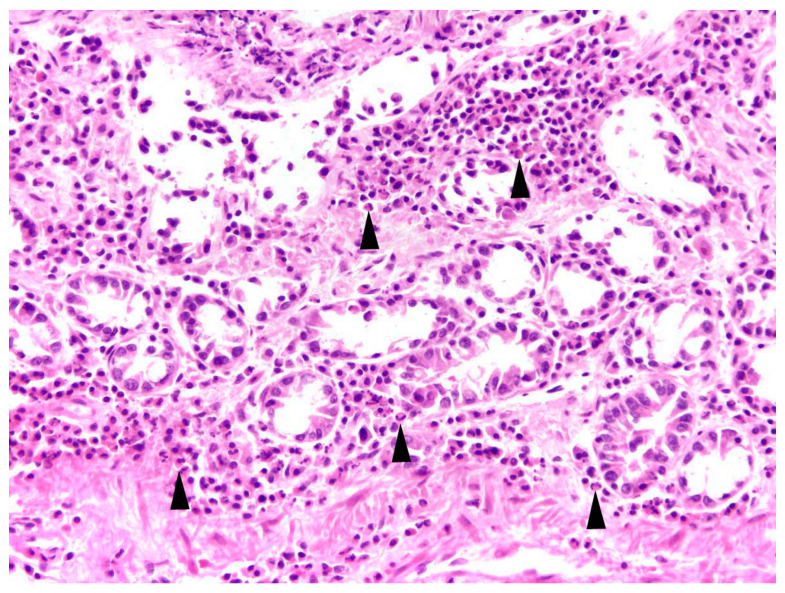
Intestine, *L*. *europaeus*; moderate lymphoplasmacytic and eosinophilic (arrowheads) enteritis. HE, 200×.

**Figure 9 pathogens-12-01317-f009:**
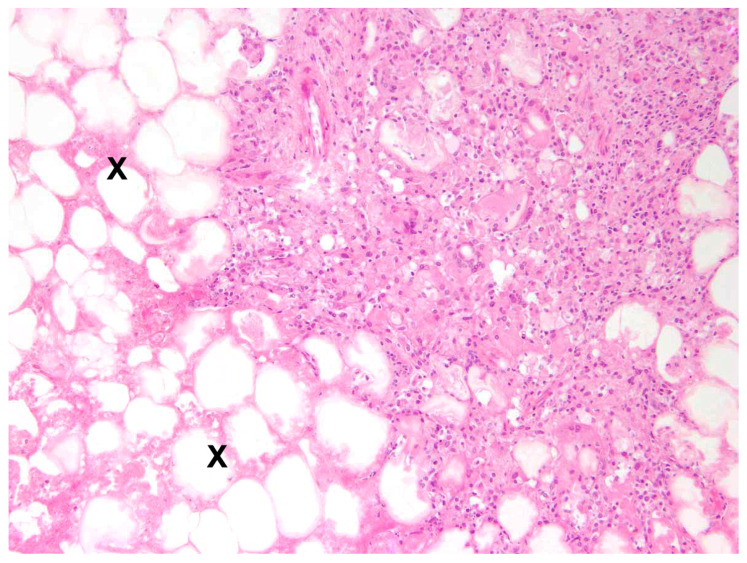
Retroperitoneal fat tissue, *L*. *europaeus*; severe granulomatous to necrotising steatitis with extensive necrosis (X) and adjacent infiltration of macrophages and multinucleated giant cells. HE, 100×.

**Figure 10 pathogens-12-01317-f010:**
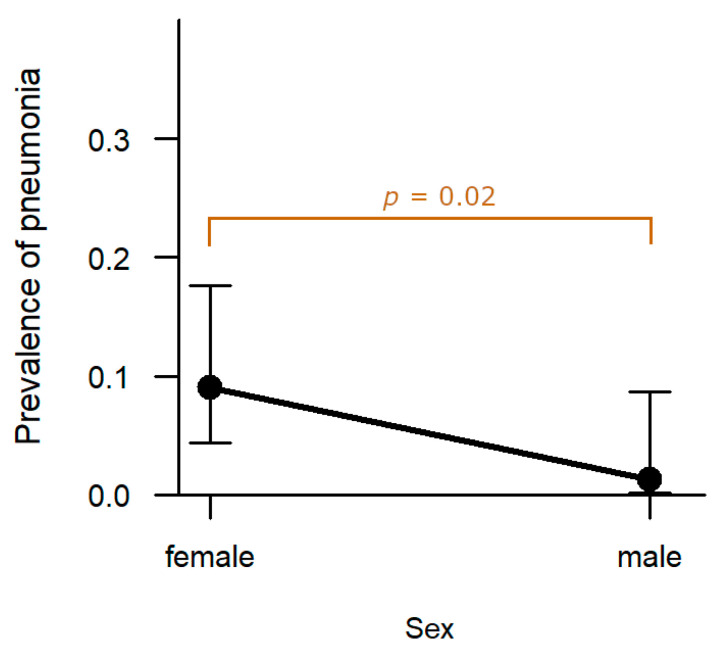
Effect of sex on the prevalence of pneumonia. Confidence bars show the 95% confidence intervals around the estimates obtained through logistic regression. Significantly more females were diagnosed with pneumonia compared to males (*p* = 0.02).

**Figure 11 pathogens-12-01317-f011:**
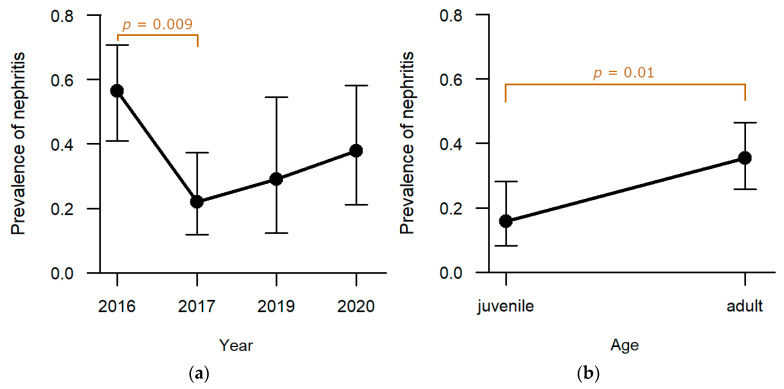
Effect of (**a**) sampling year and (**b**) age category on the prevalence of nephritis. Confidence bars show the 95% confidence intervals around the estimates obtained through logistic regression. Significant effects were detected for sampling years (2019 > 2017) and age categories (adults > juveniles) and are highlighted with square brackets and *p*-values in the graphs.

**Figure 12 pathogens-12-01317-f012:**
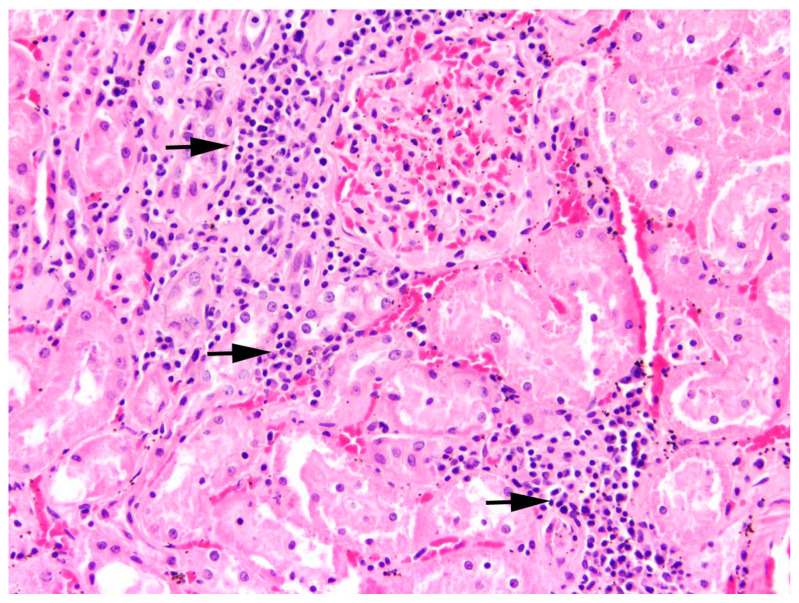
Kidney, *L*. *europaeus*; multifocal mild to moderate lymphoplasmacytic (arrows) interstitial nephritis. HE, 200×.

**Figure 13 pathogens-12-01317-f013:**
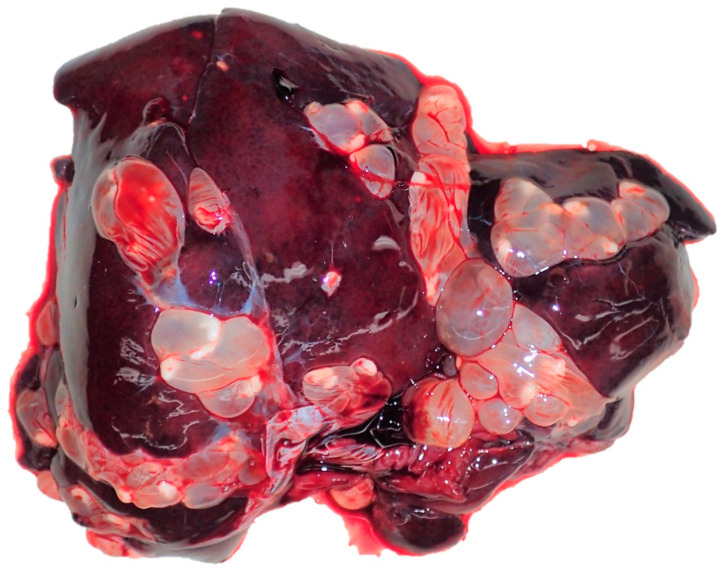
Liver, *L*. *europaeus*; severe cysticercosis (*T. pisiformis*) with multifocal cysts in the serosa of the liver of an adult male hare from Friedrichskoog.

**Figure 14 pathogens-12-01317-f014:**
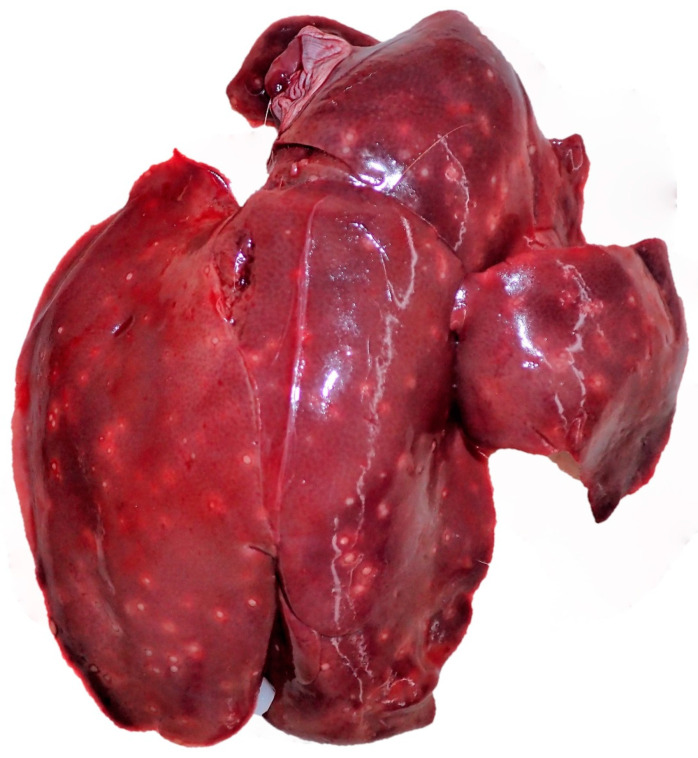
Liver, *L*. *europaeus*; multifocal bright and round spots in the liver parenchyma of an adult female hare from Elpersbüttel, infected with *B. suis* Biovar 2. The macroscopic alterations are suggestive of a granulomatous to necrotising hepatitis.

**Table 1 pathogens-12-01317-t001:** Sex and age distribution of sampled hares for hunting grounds and years. J: juvenile; A: adult; M: male; F: female. Although overall numbers of sampled animals were comparable for the different hunting grounds and sampling years, the age and sex composition differed widely between individual hunting grounds and sampling years. In total, adults outnumbered juveniles, but in sampling year 2019, an increase in juvenile-to-adult ratio (J/A) is noticeable in all hunting grounds.

	2016	2017	2019	2020	Overall
**Tetenbüll**	17	17	9	9	52
Adult	10	14	5	7	36
Female	3	7	2	5	17
Male	7	7	3	2	19
Juvenile	7	3	4	2	16
Female	4	1	2	0	7
Male	3	2	2	2	9
J/A	0.70	0.21	0.80	0.29	0.44
M/F	1.43	1.13	1.25	0.80	1.17
**Elpersbüttel**	16	14	9	9	48
Adult	9	9	2	7	27
Female	3	5	2	4	14
Male	6	4	0	3	13
Juvenile	7	5	7	2	21
Female	4	4	5	0	13
Male	3	1	2	2	8
J/A	0.78	0.56	3.50	0.29	0.78
M/F	1.29	0.56	0.29	1.25	0.78
**Friedrichskoog**	20	17	9	9	55
Adult	15	13	3	7	38
Female	7	7	3	3	20
Male	8	6	0	4	18
Juvenile	5	4	6	2	17
Female	2	2	2	1	7
Male	3	2	4	1	10
J/A	0.33	0.31	2.00	0.29	0.45
M/F	1.22	0.89	0.80	1.25	1.04
**Overall**	**53**	**48**	**27**	**27**	**155**
**J/A**	**0.56**	**0.33**	**1.70**	**0.29**	**0.53**
**M/F**	**1.30**	**0.85**	**0.69**	**1.08**	**0.99**

**Table 2 pathogens-12-01317-t002:** Chosen relevant pathomorphological findings of *L*. *europaeus* (n = 155) and their local distribution.

Pathomorphological Findings		Tetenbüll	Elpersbüttel	Friedrichskoog	Total	Total %
**Alimentary system**						
Liver and biliary tract	Hepatitis	34 (65.4%)	22 (45.8%)	30 (54.5%)	86	55.5
Intestine	Enteritis	11 (21.2%)	6 (12.5%)	4 (7.3%)	21	13.5
	Parasites in intestine	34 (65.4%)	27 (56.3%)	38 (69.1%)	99	63.9
**Abdominal and thoracic cavities**						
Peritoneum	Steatitis	11 (21.2%)	12 (25%)	13 (23.6%)	36	23.2
**Respiratory system**						
Lung	Pneumonia	2 (3.8%)	3 (6.3%)	3 (5.5%)	8	5.2
**Urinary and genital system**						
Kidneys	Nephritis	15 (28.8%)	15 (31.3%)	18 (32.7%)	48	31.0

**Table 3 pathogens-12-01317-t003:** Prevalence of detected Coccidia species (*Eimeria* (*E*.) spp.).

Coccidia Species (*Eimeria* spp.)	Total
*E. gantieri* (*Aoutil and Landau, 2005*)	19
*E. grease* (*Aoutil and Landau, 2005*)	16
*E. leporis leporis (Nieschulz, 1923)*	15
*E. stiedai (Lindemann, 1865)*	14
*E. leporis brevis* (*Aoutil and Landau, 2005*)	10
*E. deharoi deharoi* (*Aoutil and Landau, 2005*)	10
*E. nicolegerae* (*Aoutil and Landau, 2005*)	9
*E. cabareti* (*Aoutil and Landau, 2005*)	7
*E. orbiculata* (*Lucas, Laroche and Durand, 1959*)	5
*E. audubonii* (*Duszynski and Marquardt, 1969*)	4
*E. rotonda* (*Aoutil and Landau, 2005*)	3
*E. tailliezi* (*Aoutil and Landau, 2005*)	3
*E. macrosculpta* (*Sugar, 1979*)	3
*E. bainae* (*Aoutil and Landau, 2005*)	1
*E. coquelinae* (*Aoutil and Landau, 2005*)	1
*E. europaea (Pellerdy, 1956)*	1

## Data Availability

The data presented in this study are available in this published article and its Appendix A.

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
