# Peer review of "Pathomorphological Findings and Infectious Diseases in Selected European Brown Hare (Lepus europaeus Pallas, 1778) Populations from Schleswig-Holstein, Germany"

_pathogens, 2023, doi:10.3390/pathogens12111317_

Round 1

Reviewer 1 Report

Comments and Suggestions for Authors

Pathomorphological findings and infectious diseases in selected populations of hunted free-ranging European brown hares (Lepus europaeus) from Schleswig-Holstein, Germany, and their variability in location, year, age and sex

The title should be shorter, it's too long and already contains the answer to the paper.

The keywords should be different from the title.

The conclusion in the abstract: Individual population structures such as age, gender, location and time should be considered as relevant influencing factors in future studies. They are weak and don't add much.

The titles of the figures and tables should be a little more self-explanatory.

The aim of this study was to investigate whether and how infectious diseases and  pathomorphological findings in Schleswig-Holstein are influenced by certain variables,  based on the investigation of hunted animals, to address the potential role of infectious  diseases in the different development of brown hare populations in this German federal  state.

This is a case study. The study is limited and it might be better to continue collecting data, as the number of participants (155) is still small and the study area is very restricted. Some other tools or methods could have been used to better describe the organisms found.

Author Response

Response to Reviewer Comments

For research article: “Pathomorphological Findings and Infectious Diseases in Selected Populations of Hunted Free-Ranging European Brown Hares (Lepus europaeus) from Schleswig-Holstein, Germany, and their Variability in Location, Year, Age and Sex

Response to Reviewer 1:

1) Title. The title should be shorter, it's too long and already contains the answer to the paper. “Pathomorphological findings and infectious diseases in selected populations of hunted free-ranging European brown hares (Lepus europaeus) from Schleswig-Holstein, Germany, and their variability in location, year, age and sex
We shortened and changed the title to: “Pathomorphological findings and infectious diseases in selected European brown hare (Lepus europaeus Pallas, 1778) populations from Schleswig-Holstein, Germany”.
2) Keywords. The keywords should be different from the title.
Thank you very much for this hint. We now reduced and specified the keywords to the main content as following: “Lepus europaeus; hepatitis; steatitis; nephritis; EBHSV; Eimeria spp.; Trichostrongylus spp.; Graphidium strigosum”.
3) Abstract – Page 1, Lines 41-42. The conclusion in the abstract: “Individual population structures such as age, gender, location and time should be considered as relevant influencing factors in future studies.” They are weak and don't add much.
We agree with your comment and changed the last two sentences of the abstract (Lines 38-41): “This study revealed a low prevalence of infectious pathogens, but a high prevalence of chronic inflammations of unknown origin in the tested brown hare populations. Overall, our results indicate a rather minor importance of infectious diseases for observed population dynamics of analysed hare populations in Schleswig-Holstein.”
4) Figures and tables. The titles of the figures and tables should be a little more self-explanatory.
Thank you for pointing this out. We changed and specified the titles of the following figures and tables to be more self-explanatory:
Figure 1. Development of population densities for Lepus europaeus in Schleswig-Holstein from 1995 to 2022 based on spotlight counts in spring with dynamic, but different population densities for whole Schleswig-Holstein and different natural regions (Geest, Downs, Marshland) (adjusted from Wildtierkataster Schleswig-Holstein [12]).
Figure 2. Locations of the three sampled hunting grounds (triangles). From north to south: Tetenbüll, Elpersbüttel, Friedrichskoog.
Figure 3. Effect of a) sampling year and b) age category on the prevalence of hepatitis. Confidence bars give the 95% confidence intervals around the estimates obtained through logistic regression. The effect of sampling year is depicted for adults (Fig. 3a), and the effect of age category is depicted for a weighed average across the sampling year effects with weights proportional to the number of samples obtained in each year (Fig. 3b). Significant differences between sampling years and age categories are illustrated individually by square brackets including the p-value and show a significant decrease of prevalence from 2017 to 2019 and 2020 as well as a higher prevalence of hepatitis in adults.
Figure 6. Effect of age and sex on the prevalence of parasites in the intestine. Confidence bars give the 95% confidence intervals around the estimates obtained through logistic regression. A significant difference was only detected within males between juveniles and adults and illustrated by a square bracket including the p-value.
Figure 7. Effect of a) hunting ground and b) sampling year on the prevalence of enteritis. Confidence bars give the 95% confidence intervals around the estimates obtained through logistic regression. A significant effect of sampling years on prevalence of enteritis was detected in likelihood ratio tests, but post-hoc analyses did not reveal between which years this difference was present. In 2019, no enteritis was diagnosed.
Figure 10. Effect of sex on the prevalence of pneumonia. Confidence bars give the 95% confidence intervals around the estimates obtained through logistic regression. Significantly more females were diagnosed with pneumonia compared to males (p = 0.02).
Figure 11. Effect of a) sampling year and b) age category on the prevalence of nephritis. Confidence bars give the 95% confidence intervals around the estimates obtained through logistic regression. Significant effects were detected for sampling years (2019 > 2017) and age categories (adults > juveniles) and are highlighted with square brackets and p-values in the graphs.
Figure 13. Liver, L. europaeus; severe cysticercosis (T. pisiformis) with multifocal cysts in the serosa of the liver of an adult male hare from Friedrichskoog.
Figure 14. Liver, L. europaeus; multifocal bright and round spots in the liver parenchyma of an adult female hare from Elpersbüttel, infected with B. suis biovar 2. The macroscopic alterations are suggestive of a granulomatous to necrotising hepatitis.
Table 1. Sex and age distribution of sampled hares for hunting grounds and years. J: juvenile; A: adult; M: male; F: female. Although overall numbers of sampled animals were comparable for the different hunting grounds and sampling years, the age and sex composition differed partly widely between individual hunting grounds and sampling years. In total, adults outnumbered juveniles, but in sampling year 2019 an increase of juvenile-to-adult ratio (J/A) is noticeable in all hunting grounds.
Table 3. Prevalences of detected Coccidia species (Eimeria (E.) spp.).
5) Introduction - Page 3, Lines 81-85. This is a case study. The study is limited and it might be better to continue collecting data, as the number of participants (155) is still small and the study area is very restricted. Some other tools or methods could have been used to better describe the organisms found.
The aim of this study was to investigate whether and how infectious diseases and pathomorphological findings in Schleswig-Holstein are influenced by certain variables, based on the investigation of hunted animals, to address the potential role of infectious diseases in the different development of brown hare populations in this German federal state.”
You are totally right; the sample size and the regional restrictions limit the possibility for generalisation of detected results. This initial study was unfortunately only financed until 2020, but we would like to continue and implant a continuous monitoring program. For this study, we changed the sentence you refer to, with a more local statement (Lines 84-88): “The aim of this study was to investigate whether and how infectious diseases and pathomorphological findings in different locations of Schleswig-Holstein are influenced, based on the investigation of hunted animals, to address the potential role of infectious diseases in the different developments of brown hare populations in this German federal state.

Reviewer 2 Report

Comments and Suggestions for Authors

This study deals with the analysis of infectious diseases and pathomorphological cases in the European brown hare Lepus europaeus from selected populations in Schleswig-Holstein (Germany). The European brown hare is an important game animal species with high abundance. The authors tried to assess the impact of infectious diseases and the presence of pathomorphology on hare population dynamics. The manuscript undoubtedly meets the goals and objectives of the journal Pathogens and can be published.

I have some remarks about this manuscript.

Authors need to tidy up the links to figures and tables in the article. Both figures and tables should be placed immediately after the first mention of them in the text.

Figure 1 would be more appropriate to include in the section Discussion. Typically, the Introduction does not contain figures or tables.

According International Code of Zoological Nomenclature (ICZN) at the first mention of species and genus (as in lines 50, 81, 82, etc.) its full Latin name with the author and year of description should be given; in relation all species of animals and their parasites (for example, Lepus europaeus Pallas, 1778); Yersinia pseudotuberculosis (Pfeiffer 1889), Pasteurella multocida Trevisan 1887, Francisella tularensis (McCoy and Chapin 1912), Mosgovoyia pectinata (Goeze, 1782), etc.). This remark also applies to Tables 3, S4.

And for the main “hero” of the article, at the first mention, you can give the names of the family and order – Lepus europaeus Pallas, 1778 (Lagomorpha Leporidae).

On subsequent mentions, the generic name is abbreviated (as in line 92 – L. europaeus). And try not to use common names in scientific articles, use only Latin names. Common names can be mentioned once along with the Latin name, and then only the Latin J

That's why – …sampled Lepus europaeus… (line 239)

In case line 241 (text) – …sampled L. europaeus

Figure 4. Liver, Lepus europaeus (or Liver of Lepus europaeus) Here you can add the area where the hare was caught. The same applies to the names of Figures 5, 8, and 12.

Line 286 (Table name) – … of Lepus europaeus

Lines 443-445, 452 - Why were nematodes not identified to the species level? For example, as far as is known, the genus Graphidium is monotypic and contains only one species Graphidium strigosum (Dujardin, 1845). And for the genus Passalurus, only two species are known

Line 732-740 - Needs to be done as a Conclusion

And some minor remarks and noted errors in the text:

Line 438 and Table name – Coccidia species.

Lines 453 – Taenia pisiformis Bloch 1780, larvae (or metacestode Taenia pisiformis Bloch 1780)

Lines 466,467 - A paragraph cannot consist of one sentence. It should be attached to the paragraph above or below.

Line 481 – Second mention of species in text– T. pisiformis.

Line 644 – First mention, therefore Trichostrongylus retortaeformis (Zeder, 1800). Сorrect a misspelled word retortaeformis.

The manuscript can be published, but minor corrections are needed.

Author Response

Response to Reviewer Comments

For research article: “Pathomorphological Findings and Infectious Diseases in Selected Populations of Hunted Free-Ranging European Brown Hares (Lepus europaeus) from Schleswig-Holstein, Germany, and their Variability in Location, Year, Age and Sex

Response to Reviewer 2:
1) Figures and tables.
Authors need to tidy up the links to figures and tables in the article. Both figures and tables should be placed immediately after the first mention of them in the text.
Thank you very much for this hint. We deleted the links to figures and tables in the text and placed these closer to the first mention in the text.
Figure 1 would be more appropriate to include in the section Discussion. Typically, the Introduction does not contain figures or tables.
Thank you very much for pointing this out, and we totally agree that this is not the usual layout. But we intentionally included this figure at the beginning of the introduction section, as the displayed population courses gave rise to this study, and this visualisation helps to point out the primary research question and the aim of this study, which we wanted to include as main parts of the introduction section.

2) Nomenclature. According International Code of Zoological Nomenclature (ICZN) at the first mention of species and genus (as in lines 50, 81, 82, etc.) its full Latin name with the author and year of description should be given; in relation all species of animals and their parasites (for example, Lepus europaeus Pallas, 1778); Yersinia pseudotuberculosis (Pfeiffer 1889), Pasteurella multocida Trevisan 1887, Francisella tularensis (McCoy and Chapin 1912), Mosgovoyia pectinata (Goeze, 1782), etc.). This remark also applies to Tables 3, S4.
And for the main “hero” of the article, at the first mention, you can give the names of the family and order – Lepus europaeus Pallas, 1778 (Lagomorpha Leporidae).
On subsequent mentions, the generic name is abbreviated (as in line 92 – L. europaeus). And try not to use common names in scientific articles, use only Latin names. Common names can be mentioned once along with the Latin name, and then only the Latin
That's why – …sampled Lepus europaeus… (line 239)
In case line 241 (text) – …sampled L. europaeus
Thank you very much for your valuable comment on this. We totally agree with you and adjusted the first mention of species and genus according to the ICZN regulations. We also changed the generic name to the abbreviated version from the second mention of species and genus onwards and changed common names such as “brown hare”, “European brown hare”, etc., whenever possible and if not influencing the flow of
reading to L. europaeus. At least the word “hare” we would like to keep in this text, as, in our opinion, this is better for the flow of reading, and this is the only common species name in the text, so could be easily referred to.
3) Results - Page 10, Line 311.
Figure 4. Liver, Lepus europaeus (or Liver of Lepus europaeus) Here you can add the area where the hare was caught. The same applies to the names of Figures 5, 8, and 12.
We changed the word “hare” to “L. europaeus”.
4) Results - Page 9. Line 286 (Table name) – … of Lepus europaeus
We changed “[…] of European brown hares […]” to “[…] of L. europaeus […]”.
5) Results - Page 17, Lines 443-445, 452. Why were nematodes not identified to the species level? For example, as far as is known, the genus Graphidium is monotypic and contains only one species Graphidium strigosum (Dujardin, 1845). And for the genus Passalurus, only two species are known.

Thank you very much for this hint. Yes, you are right that Graphidium strigosum is the only possible species, so we changed it to Graphidium strigosum as you suggested. Most of the other nematode species could not be identified by egg morphology in parasitological analyses of faeces, so we were not able to identify these by species level.
6) Discussion – Line 732-740. Needs to be done as a Conclusion.
We included the conclusion part at the end of the discussion (Lines 743-751).
7) Results – Page 17, Line 428. Line 438 and Table name – Coccidia species.
We changed “coccidia” to “Coccidia” (Line 450).
8) Results - Page 17, Line 443. Taenia pisiformis Bloch 1780, larvae (or metacestode Taenia pisiformis Bloch 1780)
We changed this sentence to: “[…] and the cestodes Mosgovoyia pectinata (Goeze, 1782) (adult worm; n = 1) and Taenia (T.) pisiformis (Bloch, 1780) (metacestode; n = 1) […]” (Lines 465-466).
9) Results – Page 18, Lines 456-457. A paragraph cannot consist of one sentence. It should be attached to the paragraph above or below.
Thank you very much for your comment on this. We attached the paragraph to the one above (Lines 481-482).
10) Results – Page 17, Lines 471. Second mention of species in the text– T. pisiformis.
We changed it as suggested (Line 470) and introduced the abbreviation T. pisiformis with the first mention of Taenia pisiformis (Line 466).
11) Discussion – Page 23, Line 631. First mention, therefore Trichostrongylus retortaeformis (Zeder, 1800). Сorrect a misspelled word – retortaeformis.
Thank you very much for finding this mistake on spelling, which we corrected (Line 655). The abbreviation T. retortaeformis was already introduced in line 82-83.

Round 2

Reviewer 1 Report

Comments and Suggestions for Authors

The article has improved a lot.